# Acceptability of fixed-dose combination treatments for hypertension in Kenya: A qualitative study using the Theoretical Framework of Acceptability

**Daniel Mbuthia**[1], **Ruth Willis**[2], **Mary Gichagua**[3], **Jacinta Nzinga**[1,4], **Peter Mugo**[1†], **Adrianna Murphy**[2]*

1 Health Economics Research Unit, KEMRI-Wellcome Trust Research Programme, Nairobi, Kenya, 2 Department of Health Services Research and Policy and Centre for Global Chronic Conditions, London School of Hygiene and Tropical Medicine, London, United Kingdom, 3 Department of Health, Kiambu, Kenya, 4 Department of International Public Health, Liverpool School of Tropical Medicine, Liverpool, United Kingdom

† Deceased
* Adrianna.Murphy@lshtm.ac.uk

## Abstract

Fixed-dose combinations (FDCs) – 2-3 anti-hypertensive medications in a single pill - have the potential to improve hypertension treatment and outcomes. Yet, they are not widely implemented. Factors undermining implementation remain unknown, particularly in sub–Saharan Africa, where hypertension is a major cause of disease burden and is poorly controlled. This study explored the acceptability of FDCs among patients, caregivers, and healthcare workers. We conducted semi-structured in-depth interviews with 58 participants from four purposively selected health facilities in Kiambu county, Kenya. Data were analyzed using an iterative thematic analysis approach, guided by the Theoretical Framework of Acceptability. Our findings indicate that FDCs are potentially acceptable to all participant groups. Acceptability is supported by the perception of FDCs as a means of reducing treatment burden (for patients and healthcare workers) and improving treatment adherence, and by patients' deferral to and trust in healthcare workers. However, acceptability among healthcare workers may be undermined by variable levels of knowledge about FDCs, concerns about FDCs as an "inflexible" treatment that does not allow dose titration or identifying causes of side effects, and concerns about inconsistent availability and affordability of FDCs in Kenya. To enhance acceptability and implementation of FDCs for hypertension treatment in Kenya, it is crucial to strengthen the capacity of all healthcare worker cadres to appropriately prescribe, inform patients about, and support adherence to FDCs. These efforts must align with broader initiatives to address upstream health system factors such as poor availability and affordability.

**Data availability statement:** The article contains de-identified excerpts from the qualitative data we collected and synthesized. De-identified individual participant data that underlie the results reported in this article have been deposited in a data repository and are available for controlled access via LSHTM Data Compass (https://doi.org/10.17037/DATA.00004498).

**Funding:** We acknowledge UKRI for funding this research study through the UKRI Future Research Leader Fellowship grant awarded to Dr Adrianna Murphy by the UK Research and Innovation fund, grant number MR/T042508/1. The funding sources had no role in the study design, writing of the report, or in the decision to submit the manuscript for publication.

**Competing interests:** The authors have declared that no competing interests exist.

**Abbreviations:** CHPs: Community health promoters; CVD, Cardiovascular diseases; EML, Essential medicines list; FDCs, Fixed dose combinations; LMICs, Low- and middle-income countries; NCDs, Non communicable diseases; NHIF, National Health Insurance Fund; TFA, Theoretical Framework of Acceptability.

# Background

Cardiovascular disease (CVD) is the leading cause of death globally, accounting for almost 18 million deaths annually [1]. Of these, over 10 million CVD deaths are due to uncontrolled hypertension (blood pressure ≥ 140/90 mm Hg). Most of these deaths occur in low- and middle-income countries (LMICs) [1] and are in part due to low rates of treatment. In Kenya, for example, the prevalence of hypertension among people aged 18–69 years is estimated at 24.5% [2]. However, only 15.6% -29.4% of these patients are diagnosed. Among those diagnosed, only 6.5% are on antihypertensive medication, and just 12.5% - 51.7% of those on medication have their blood pressure controlled [3,4]. The gap in treatment of hypertension has been attributed to inconsistent availability and affordability of antihypertensives [5], poor health literacy among patients and poor adherence to prescribed treatment [6,7] the high number and diversity of anti-hypertensive medication treatment options, which can make treatment complex [8], and 'therapeutic inertia', or the failure of physicians to initiate or intensify therapy when BP remains elevated [9] and the need for healthcare workers to be adequately trained to manage hypertension [10]. Despite evidence that 75-80% of hypertensive patients require multiple classes of drugs to effectively control their blood pressure, many receive monotherapy (one drug) [11].

## Fixed-dose combination treatments: A potential solution for bridging the treatment gap

Fixed-dose combination (FDC) treatments – a combination of 2-3 anti-hypertensive medications in a single pill – offer one potential tool to address the gap in hypertension treatment in LMIC settings. FDCs can enhance patient compliance by reducing the pill burden, making it easier for patients to adhere to their treatment regimen compared to taking multiple single-molecule pills [12]. FDCs have been shown to have other benefits including faster achievement of blood pressure targets [13], fewer CVD events [14], reductions in therapeutic inertia [15,16], and reduction in healthcare costs [17]. In 2019, the World Health Organization (WHO) included FDCs in their Model Essential Medicines List and released a new guideline in 2021 recommending use of FDCs for treatment of hypertension [18]. In Kenya, FDCs for hypertension are authorized for marketing and there are nine FDCs included in the 2023 Kenya National Essential Medicines List [EML] (amlodipine + hydrochlorothiazide, amlodipine + indapamide, perindopril + amlodipine, perindopril + amlodipine + indapamide, losartan + hydrochlorothiazide, lisinopril + hydrochlorothiazide, telmisartan + amlodipine, telmisartan + hydrochlorothiazide and telmisartan + amlodipine + hydrochlorothiazide) [19] and FDCs are recommended in the treatment guidelines for hypertension [20]. However, uptake of FDCs in LMICs, including in Kenya, has been slow [21].

To inform the development of a strategy to improve implementation of FDCs for treatment of hypertension in Kenya, we aimed to evaluate the acceptability of FDCs for patients with hypertension, their caregivers, and healthcare workers.

# Methods

## Conceptual framework: Theoretical Framework of Acceptability

To conceptualize acceptability of FDC treatments for hypertension, we used the Theoretical Framework of Acceptability (TFA) [22]. TFA has been applied in implementation research across a range of settings and health domains, including for hypertension and CVD management in LMIC settings [23,24]. TFA identifies seven component constructs contributing to acceptability of healthcare interventions. These include: Affective attitude (How an individual feels about the intervention); Burden (The perceived amount of effort that is required to participate in the intervention); Ethicality (The extent to which the intervention has good fit

with an individual's value system); Intervention coherence (The extent to which the individual understands the intervention and how it works); Opportunity costs (The extent to which benefits, values, or profits must be given up to engage in the intervention); Perceived effectiveness (The extent to which the intervention is perceived as likely to achieve its purpose); and Self-efficacy (The individuals' confidence that they can perform the behaviour(s) required to participate in the intervention). S1 Appendix presents our interpretation of each construct for acceptability of FDC treatment for hypertension, which informed our overall study design, data collection tools and analysis.

## Study design

We used an explorative qualitative study design. This design allows for in-depth exploration and understanding of a phenomenon when there is limited prior evidence - to the best of our knowledge, this is the first study exploring the acceptability of FDCs for hypertension in Kenya, and one of the few in the world. We conducted non-participant observation and semi-structured in-depth interviews in Kiambu County, Kenya. Non-participant observation was used to orientate the study team to facility operations and current practices for treatment of patients with hypertension in context, and to inform criteria for selection of participants for semi-structured interviews. Semi-structured interviews were used to explore participant experiences and attitudes in depth.

## Study setting

The study was set in one county in order to develop a robust understanding of a specific context [25] to enable identification of both locally specific factors affecting acceptability, and the categories of factors to consider in potential design and scale-up of a subsequent intervention to improve implementation of FDCs. The choice of county was informed by consideration of (i) contextual diversity, to include urban and rural populations from a range of socio-economic situations; (ii) receptiveness of county stakeholders to NCD implementation research, while not being overburdened with ongoing research projects, and (iii) practicality in relation to the operational base of the study team in Nairobi and to national policy stakeholders. Kiambu County is adjacent to Nairobi County in central Kenya, and whilst aggregated county level socio-economic indicators are above the national average, for example overall poverty rate of 20.5% (Kiambu County) compared with 38.6% (national) [26], there is diversity within the county population across 14 urban and rural sub-counties [27].

The Kenyan public health system is organized in 6 care levels. Treatment of hypertension in public hospitals is mainly delivered at primary healthcare centres (level 3) subcounty hospitals (level 4) and county referral hospitals (level 5) [18]. We collected data in four public sector facilities located in three sub-counties, purposively selected with input from county stakeholders to include a range of facility levels, urban/rural and socio-demographically diverse catchment populations. Based on these criteria, we selected one county referral hospital (level 5), one sub-county hospital (level 4), and two primary healthcare centres (level 3) serving contrasting urban and rural populations [28]. All facilities provide care to people living with hypertension, including prescription of anti-hypertensive medication, through Non Communicable Disease (NCD) or Medical Outpatient Clinics (MOPCs). Specific facilities selected and their characteristics are outlined in Table 1.

## Participant recruitment

On clinic days, with the help of a triaging nurse, the researchers prospectively and purposively selected patients living with hypertension that were attending the hypertension clinic,

**Table 1. Study facility characteristics.**

| Facility | Characteristics |
|---|---|
| Health facility 1 | Level 4 sub-county referral hospital<br>Serves mostly urban population<br>High patient volume<br>Medical officers, clinical pharmacists, clinical officers [non-physician clinicians], nurses, nutritionists |
| Health facility 2 | Level 3 primary healthcare center<br>Serves urban population<br>Low patient volume<br>1 Medical officer and 1 clinical officer |
| Health facility 3 | Level 5 county referral hospital<br>Serves both urban and rural populations<br>Very high patient volume<br>Medical consultants, medical officers, clinical pharmacists, clinical officers, nurses, nutritionists |
| Health facility 4 | Level 3 primary health center<br>Serves rural population<br>High patient volume<br>1 nurse supported by 1 community health promoter (CHP) (often referred to as community healthcare workers in other settings) and occasional locum hire of clinical officer |

drawing on patient registers to identify patients representing a range of categories relevant to experiences of hypertension treatment (age, sex, comorbidities, length of time since diagnosis, caregiver accompaniment) (Table 2). A small number of caregivers (1-2) were recruited from each facility following patient interviews, with prior permission from patients. Healthcare worker participants were purposively selected to ensure inclusion of one member from each staff cadre involved in hypertension treatment at each facility.

Eligible study participants were approached face to face or by telephone (caregivers) and invited to join the study. Participants were provided with a study information sheet [English/Swahili]. DM reviewed the information sheet with each participant and written consent was obtained from all participants. Recruitment at each facility stopped when patients from a pre-specified range of category combinations had been included, and no substantial new themes were emerging during interviews. A total of 58 participants were involved in this study (Table 3).

## Data collection

Data were collected between November 2022 and June 2023. First, we conducted non-participant observations spanning 1-3 days in each facility, (DM, PM, RW) structured by a checklist designed to support familiarization with medication procurement and dispensing [S1 Text]. Observational data were recorded in fieldnotes, subsequently summarized, and cross-checked with facility staff. Medical consultations were not observed.

Second, we conducted in-depth interviews using semi-structured topic guides [S1 Text] informed by the TFA constructs applied to FDCs for hypertension [S1 Appendix]. The interview guide for patients and caregivers focused on experiences with treatment, support in managing hypertension, the process of obtaining medication, and familiarity with and perceptions about FDCs. The interview guide for healthcare workers focused on the process of diagnosing and treating hypertension and experiences of and views around use of FDCs to treat hypertension. Interviews were conducted in Swahili/English depending on the participant's preference. Patient and caregiver interviews were conducted by DM, mainly in Swahili. Interviews with

**Table 2. Patient and caregiver characteristics.**

| Patients | | Number (total n=24) |
|---|---|---|
| Gender | Female | 14 |
| | Male | 10 |
| Age (years) | <40 | 2 |
| | 41-50 | 2 |
| | 51-60 | 9 |
| | 61-70 | 6 |
| | >71 | 5 |
| Mean Age | 60 years | |
| Education level | None | 1 |
| | Primary | 6 |
| | Secondary | 15 |
| | Tertiary | 2 |
| Comorbidity | Hypertension | 13 |
| | Hypertension + Diabetes | 11 |
| National health insurance fund (NHIF) membership | Active NHIF membership | 12 |
| **Caregivers** | | Number (total n=7) |
| Gender | Female | 5 |
| | Male | 2 |
| Age (years) | 18-40 | 4 |
| | >40 | 3 |
| Mean Age | 41 years | |
| Education level | Primary | 3 |
| | Secondary | 3 |
| | Tertiary | 1 |
| Patient Comorbidity | Hypertension | 2 |
| | Hypertension + Diabetes | 5 |

**Table 3. Study participants across levels of care.**

| Participant Role | Level 3 Primary Health Centres | Level 4 & 5 Hospitals | Total |
|---|---|---|---|
| Patients | 12 | 12 | 24 |
| Caregivers | 2 | 5 | 7 |
| Community Health Promotorss | 4 | 0 | 4 |
| Nurses | 2 | 2 | 4 |
| Pharmacists/Pharmaceutical Technologists | 3 | 3 | 6 |
| Community Pharmacists at private sector pharmacies local to the public sector study facility | 1 | 3 | 4 |
| Clinical Officers/Medical DoctorsMedical Consultants | 2 | 7 | 9 |
| Total | 26 | 32 | 58 |

healthcare workers were conducted by DM and RW (social scientists experienced in qualitative research) in English or Swahili. Interviews were conducted in person in a private room at each facility, with only the researcher(s) and participant present. Interviews lasted 15-84 minutes and were audio recorded with participants' consent. During data collection and analysis, reflective meetings were held with the broader research team, composed of researchers with

diverse backgrounds and experience (public health, medicine, pharmacy) to discuss interpretation of emergent findings and iteratively inform data collection. Additionally, a stakeholder workshop comprised of study participants, county and national level stakeholders was held to discuss the preliminary findings and identify priority themes for further exploration in subsequent analysis. Audio data were transcribed verbatim and translated into English where necessary. Transcripts were checked for accuracy, translations reviewed, and transcripts imported into NVIVO 12 (QSR International, Australia) for analysis.

### Data analysis

We used a hybrid approach of inductive and deductive thematic analysis [29]. First, transcripts were read and discussed, and open coding was used to inductively identify initial themes of relevance to the research aim. Initial themes were then compared with TFA constructs and used to develop a coding framework which was applied across the transcripts, and adapted iteratively as needed. Two researchers independently coded a set of five transcripts to ensure inter-coder reliability. Themes were then charted across the seven TFA constructs. The Standards for Reporting Qualitative Research (SRQR) guidelines [30] were followed in our reporting (S1 Checklist).

### Inclusivity in global research

Additional information regarding the ethical, cultural, and scientific considerations specific to inclusivity in global research is included in the supporting information (S2 Checklist)

### Results

In our research we found that the only FDC available in the public sector in Kiambu County during the study period was a dual therapy FDC of losartan (an angiotensin receptor blocker (ARB)) + hydrochlorothiazide (a diuretic). Of the 24 patients interviewed, seven were currently prescribed this FDC, while the rest were prescribed separate pills. Therefore, data from patient interviews include both anticipated (n=17) and experienced (n=7) cognitive and emotional responses to FDCs for hypertension. We present key themes affecting acceptability of FDCs for patients and caregivers, and then for healthcare workers, according to the TFA construct to which they relate. Only TFA constructs for which there was evidence from our study of key themes related to acceptability are presented in our results. Fig 1 gives an overview of findings.

### Acceptability of FDC treatment for hypertension for patients and caregivers

**Affective attitude: FDCs as a means of reducing treatment burden.** Data from patient and caregiver interviews suggested a positive perception of FDCs as a means of reducing overall burden associated with taking FDCs compared with taking multiple single-molecule pills. The perception of FDCs as a means of reducing treatment burden was related to the reduction in the burden of obtaining and taking multiple pills, including the time and effort needed to obtain separate pills if they were not available at the public pharmacy, the cognitive work of remembering the correct daily medication schedule and planning ahead to take medication supplies if travelling, the physical discomfort of taking several tablets, and cost. For example:

> *"It would be better than what I'm getting now, separate pills. If it were possible to combine all these medicines together, that would be great… it would save me time of having to look for the other medicines [individual medicines]."* [Patient_17]

Key constructs of the Theoretical Framework of Acceptability in acceptability of FDCs for hypertension

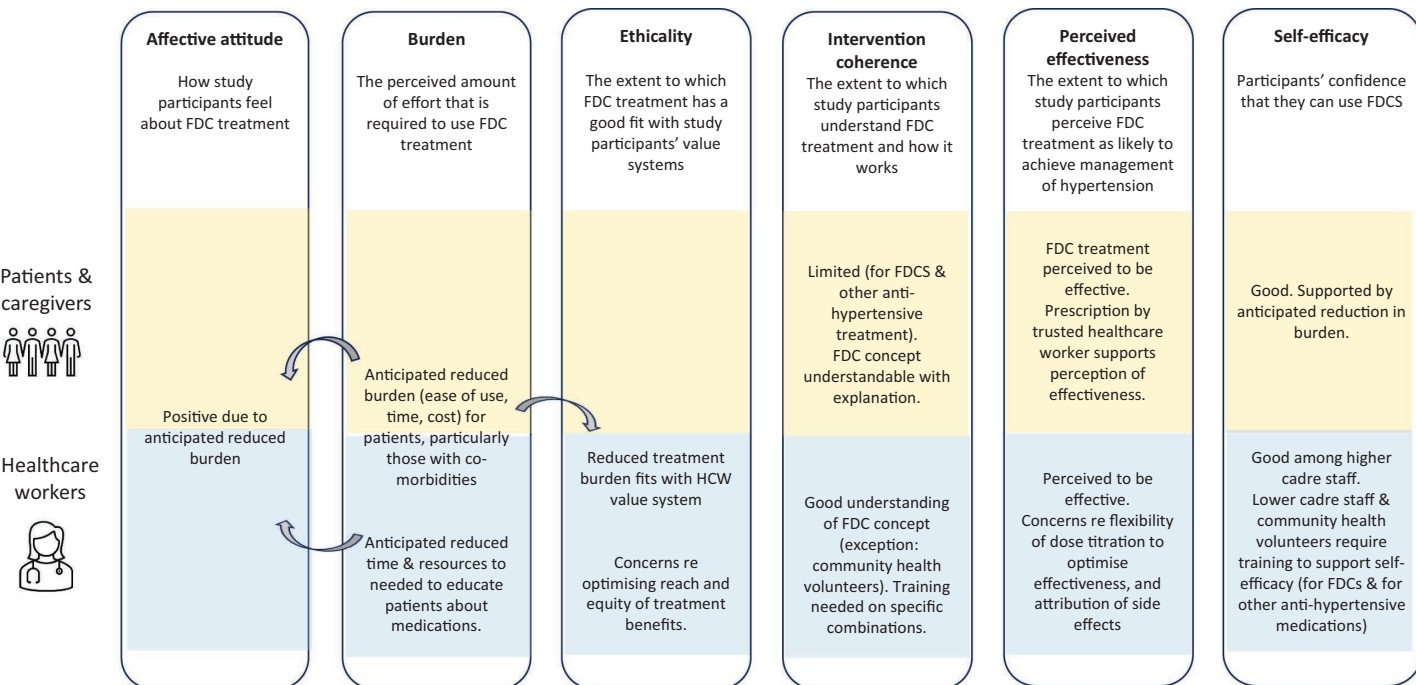

**Fig 1. Theoretical framework of acceptability constructs.**

A combined pill was seen as more convenient - '*you take it once and off you go*' [Patient_01] - and easier than taking multiple tablets that can feel '*stuck*' in the throat; '*when you take one [pill] it's good because it will make the work of swallowing [medicines] easier*" [Patient_02]. It was also seen as less costly: "*buying the medicine when it's one [combined], it's not like buying three [separate pills]. One is much cheaper to buy than buying three*' [Caregiver_07]. The potential role of FDCs in reducing the burden of taking many pills was particularly highlighted by patients with comorbidities. For example, a patient with hypertension, diabetes and HIV noted that when taking multiple medications for different conditions '*you are in trouble because you don't know if you should take them all at once*', and that if taking an FDC for hypertension '*the luggage of taking a lot of medicine has been reduced*' [Patient_11]. A patient with hypertension and diabetes explained misgivings about ingesting large volumes of '*chemicals*' when taking multiple pills, and expressed preference for FDC treatment to reduce the volume of chemicals introduced to the body through separate pills, echoing concerns of several patients that taking multiple medications could be '*harmful*' to the body.

**Intervention coherence: Low health literacy undermining intervention coherence.** Overall, the extent to which patients and caregivers understood FDCs, or hypertension treatment generally, was low. This is in part due to a lack of information about the concept of FDCs (with the exception of a small number of patients who had prior experience of FDCs in treatment of other conditions such as HIV or diabetes). Some patients taking the losartan + hydrochlorothiazide combination were unaware that their medication was an FDC. Most patients were not familiar with the medication they were currently taking, for example knowing the name (e.g., losartan) or class (e.g., ARB). Only one patient could describe in simple terms how their antihypertensive medication

worked. Some expressed misunderstandings of why an FDC would be effective, suggesting that FDCs create a synergistic effect due to the drugs being combined, for example: "*I think it can work faster when it is one pill than the other ones [separate pills].*" *[Patient_24].* Discussing prospective acceptability, patients emphasized the importance of the concept of combined medications being clearly explained so that patients understood why the number of medicines in their prescription was reduced in order to address concerns over missing important medication components, for example "*it will be best to explain to the patients that the drugs have been combined into one pill because they must have questions…why would I take one instead of three tablets as I was taking?*" *[Patient_13]*

**Deferral of decision making to healthcare workers.** Our interviews with patients and caregivers revealed a key theme affecting FDC acceptability that did not fit neatly under any of the TFA domains - a very strong trust in and deferral to healthcare workers. The interviews suggested a belief that knowledge and decisions about medications are the responsibility of healthcare workers and should not be questioned:

> "*A patient cannot do research. Their doctor does the research and tells them that if they take this medication, it has no side effects or something of the sort. So, we [patients] wait for the decision from the doctor*" *[Patient_19]*

Some patients also noted that they trusted government facility staff to prioritize patients' best interests in prescription decisions, unlike in the private sector where they felt staff may prioritize financial profits.

**Perceived effectiveness: No concerns about effectiveness of FDC treatment.** Patients and caregivers did not express concerns about effectiveness of FDCs compared with single molecules, instead anticipating that their doctor would prescribe effective medication based on knowledge of their condition, reinforcing the importance of trust in healthcare workers to make decisions about medications.

**Self-efficacy: Simplification of treatment regimen supports ease of use.** Patients' self-efficacy, their confidence that they can use FDC treatment, was related to an anticipated reduction in burdens making FDCs easier to use than separate single medications, and therefore supporting self-efficacy. For example, a patient explained that FDC treatment

> "*will reduce the stress of thinking about how you shall take the medicines*" *[Patient_11].*

## Acceptability of FDC treatment for hypertension for healthcare workers

**Affective attitude: FDCs as a means of reducing treatment burden.** Healthcare workers' affective attitude - how they felt about FDCs - was a related to perceived reduced burden associated with FDCs compared with multiple single-molecule pills, although healthcare workers identified a wider range of drivers of this burden, considering both burden for the patients and for themselves and the health system. For instance, FDCs were perceived as a treatment approach that could support patient treatment adherence by reducing the likelihood that one individual medication would be unavailable:

> "*if we are to be able to package all… drugs in one pill… then I think patients would like it…the compliance is much better and the other side [with separate pills] you know sometimes this drug is over [supply run out] then they only have to use this one…*" *[Medical Officer_01]*

Reduction in the number of pills required was particularly important when healthcare workers considered patients with co-morbidities. In addition to reducing practical burdens associated with taking multiple pills, FDCs were seen to confer a psychological benefit to patients, as patients associate taking many pills with a condition being *'serious and it's like they are going to die'* however *'when it's few medicines, they feel like their condition is not that serious.' [CHP_01].* Reducing the number of pills through introduction of combined losartan + hydrochlorothiazide was described as having been a *'relief'* for patients, who a nurse predicted would welcome other combinations as *'they always look forward to taking fewer drugs' [Nurse_18].*

Healthcare workers also perceived FDCs as something that would reduce their own burden by reducing the time and resources needed to educate patients about their treatment when prescribing or dispensing medication.

"*it's even easier for me as a person who is dispensing to explain a drug when it's in fixed dose combination as opposed to explaining like three different drugs to a patient who is not getting it" [Pharmacist_07]*

**Intervention coherence: Comparison with FDCs for HIV treatment.** The concept of FDCs was well understood by all cadres of healthcare workers except by CHPs. For the CHPs interviewed, intervention coherence was similar to that for patients and caregivers. CHPs were unfamiliar with the concept of FDCs, but demonstrated understanding after neutral explanation by the interviewer, showing potential for intervention coherence if educational support is provided. CHPs noted a similarity with FDCs used in the treatment of HIV, suggesting that familiarity with FDCs for HIV might inform intervention coherence, and thus acceptability, for FDCs for hypertension. For example:

*"Just like they did with HIV, they used to prescribe so many medicines, these days they give out one, which is good." [CHP_02]*

**Ethicality: FDCs as cause of inequality in access to or affordability of treatment.** Healthcare workers' value systems prioritised equitable distribution of treatment benefits to as many patients as possible. While the anticipated reduction in burden of FDCs fits well with this value system, FDCs were often perceived to be costlier, and possibly not suitable for all patients, raising ethical concerns regarding how best to distribute limited financial resources to benefit the most patients:

" *[with] a fixed dose, you find some are taking this [molecule], some are not… the best thing is to have them separate [separate pills] so that I can serve these two branches of clients at the same time." [Pharmacist_17]*

Further, healthcare workers felt ethically obliged to prescribe consistently available and affordable medication, and therefore expressed reluctance to prescribe FDCs if doing so created financial or logistical barriers to treatment. As such, the combination of losartan+ hydrochlorothiazide- less expensive than the individual drugs as separate pills- was the only FDC procured across the four health facilities, supporting the idea that cost is an important consideration. Some healthcare workers had experienced an initiative introducing FDC of lisinopril + hydrochlorothiazide at subsidized cost in selected higher-level facilities for a fixed time period [31] and observed that the FDC became unaffordable and difficult to locate when the initiative ended, for example:

*"In the hospital [it was sold] at Ksh. 200 or 300. The same drug outside, in the chemist, would be almost Ksh. 1000. Then sustainability of that supply was not there…the prescriber prescribes and then the patient comes two, three times and they're like 'I can't find this medicine out there'…they're going to switch to a more readily available molecule"* [Pharmacist_04]

**Perceived effectiveness: FDCs as a means to improve treatment adherence, but with limitations.** Healthcare worker interview data demonstrated an overall attitude that FDC treatment would be more effective in achieving blood pressure control than separate pills. This was due to perception of FDCs as simpler and more likely to be taken as prescribed, as explained by a pharmacist:

*"They're standardized in a way… as opposed to previously having a particular pill where the patient was forced to split [it] into two, and you are not sure whether the patient is going to achieve that …the fixed dose, because they come already prepackaged, and a single tablet has that particular concentration of both drugs… It makes the plasma concentration of that particular drug…. within the required range to achieve that effect that is desired"* [Pharmacist_08]

However, the perceived effectiveness of FDCs was tempered by the perception of FDCs as "inflexible". The inflexibility, attributed to the combined nature of FDCs means that they do not allow for dose titration of individual drug components, or the identification of which individual component may be causing any side effects.

*"some are dosages issues… like the Losartan H. Maybe you need a higher dose of HCTZ [hydrochlorothiazide], but in FDC it's around 12.5mg. The second thing is concerning what we call allergies [side effects]... There are those people [patients] who come and complain… So sometimes if you have FDC, you don't know whether it is the first drug, the second drug or third drug that has an issue.* [Clinical officer_19]

**Self-efficacy: Capacity building as crucial to supporting FDC implementation.** Familiarity with FDCs varied across cadres of healthcare workers in our study. While physicians working in higher-level facilities had some experience with FDCs, in lower-level facilities, clinicians had experience only indirectly through prescriptions of patients referred from higher level facilities. Data from clinical officers and nurses, who were required to prescribe medications in lower-level facilities, highlighted the lack of awareness of hypertension medications generally, *'most of us we don't have that knowledge'* [Nurse_02], and of FDCs in particular, which may undermine levels of self-efficacy in prescribing FDCs. As described by one clinical officer:

*"so, one thing we need to do in terms of support is training! Training! Training! That is number one. You need to let everyone know that there are combinations [FDCs]. I might be knowing a few but not every other drug. You might tell me of some combination [FDCs] and I ask you, "When was it launched in Kenya? How long has it been in use?" So, we need training"* [Clinical officer_19]

## Discussion

We applied the TFA [22] to evaluate acceptability of FDC treatment, for patients with hypertension, their caregivers and healthcare workers at facilities in Kiambu County. Our results

have implications for efforts to improve implementation of FDCs, as well as for the application of the TFA to similar research.

Overall, we found that acceptability of FDCs is potentially high, driven by the perceived capacity of FDCs to reduce treatment burden for patients and healthcare workers. The perception of FDCs as a means of reducing treatment burden is supported by evidence of the positive impact on treatment adherence [32–35], and reducing therapeutic inertia [36], and should therefore be promoted by any efforts to improve uptake of FDCs in Kenya.

We also found that patients defer to healthcare workers' understanding of and decisions on treatment. Deferral of patients to healthcare workers, or trust in healthcare workers, is not unusual [37], and is likely helpful in promoting uptake of evidence-based medicine, including in relation to combination therapy for cardiovascular disease [38]. In our study, we found that deferral to healthcare workers was sometimes accompanied by misconceptions among patients about hypertension treatment (e.g., that an FDC acts on the system more quickly than individual molecules), therefore related to intervention coherence. Knowledge among patients about their medication is crucial to improving treatment adherence and reducing poor health outcomes [39,40]. Efforts to improve effective implementation of FDCs and hypertension treatment generally should consider improving patients' health literacy. Evidence shows that health literacy impacts medication adherence, especially for patients with chronic illnesses [41]. Healthcare workers should therefore be encouraged to educate patients with chronic conditions about their treatment plans including why FDCs are used, particularly for those being switched from separate pills.

The deferral of patients to healthcare workers also clearly highlights the importance of ensuring acceptability of FDCs among healthcare workers, both to promote prescription of FDCs and knowledge sharing to patients. Like previous studies on FDCs, we found that acceptability among healthcare workers is likely to be impacted by concerns about FDCs as "inflexible" (for identifying causes of side effects and dose titration) [36,42–45], intervention coherence, and feelings of self-efficacy, or capacity to appropriately prescribe and support adherence to FDCs [46–48]. Efforts to improve implementation of FDCs should therefore consider strengthening knowledge and capacity of healthcare workers of all cadres, including through dissemination of and training in clinical treatment guidelines. Training may focus on the use of FDCs with patients who have shown tolerance to constituent drugs [24,44], to reduce concerns about side effects. Strengthening capacity of lower cadre healthcare workers will be particularly important where growing integration of NCDs in primary care shifts new responsibility for treatment prescription and support to these cadres. Similar approaches have been shown to work for hypertension and other chronic conditions like HIV and TB, where building capacities of lower-level cadres has been shown to improve their knowledge and confidence in taking up these roles as well as improved patient outcomes [10,49,50]. Capacity strengthening initiatives should be continuous rather than "one-off", possibly incorporating annual refreshers or updates similar to those applied in HIV management training [51–54].

Another major driver of acceptability of FDCs among healthcare workers was consistent availability and affordability of this treatment option. While this may seem an obvious pre-requisite for implementation, key health system enablers of implementation are often overlooked by implementation trials of hypertension treatment, including of FDCs [21]. Any effort to improve awareness and capacity among patients and healthcare workers must be complemented by work to promote prioritisation of low cost generic FDCs in treatment guidelines, county procurement, and reimbursement schemes. The case for prioritisation can be supported by evidence that FDCs are more cost effective to implement compared to usual care (multiple single-molecule pills) [55], with potential cost savings to the health system attributed to better BP control and reduced hospital visits [56]. While FDCs are an important

component in improving treatment of hypertension, a comprehensive approach in managing hypertension in Kenya must address several known barriers, including improving screening and diagnosis of patients. An expert panel on control of hypertension in Africa recommended among other strategies routine and opportunistic screening of patients in every clinical encounter [57] and this can be supported through active community screening and linkage to care through community health promoters.

Finally, our study provides insights for the application of the TFA to research on FDCs or other treatment approaches for NCDs, and for future refinement of the TFA. We found the TFA enabled a granular assessment of acceptability, and highlighted inter-relationships between domains, particularly the influence of anticipated burden on other domains. The ethicality domain, defined as 'the extent to which FDC treatment has a good fit with patients' value systems' was challenging to operationalise with patients and caregivers, although it was a clearly conceptualized domain among healthcare workers reflecting on their role in delivering care. This may be because in the space of the interview, patients did not easily identify or reflect on their own value system in relation to treatment, and replacement of multiple single molecules with FDC treatment was a sufficiently similar fit to not provoke reflection or comment. A recent review of research that applied the TFA to acceptability of HIV prevention and treatment measures noted that ethicality was not reported as a distinct construct, and suggested that future refinement of the TFA reconsider the role of value systems and measurement of this dimension [58]. We suggest that ethicality remains a relevant construct but may be more readily identified by people receiving interventions where the intervention more dramatically disrupts value systems than where it has a good fit or is less 'seen'. We also found that trust in healthcare workers played a significant role in informing acceptability and was distinct from the existing TFA constructs. This theme merits exploration in future studies on treatment approaches for NCDs, and in refinement of the TFA.

## Limitations

This study was conducted in public facilities in one county in Kenya and therefore was not intended to be nationally representative of the Kenyan population. However, it includes a diverse, purposively selected participant sample of patients, caregivers, CHPs, nurses, pharmacists, clinical officers and medical doctors drawn from different levels of care, strengthening transferability.

## Conclusion

FDCs are a potentially acceptable treatment approach for hypertension in Kenya. Efforts to improve acceptability and thus implementation of FDCs in Kenya should consider improving hypertension patients and caregivers understanding of treatment and strengthening the capacity of all healthcare worker cadres to appropriately prescribe, inform about, and support adherence to FDCs. These efforts must align with work to address upstream health system factors such as poor availability and affordability, which will impede implementation. The TFA provides an appropriate framework for exploring the multifaceted nature of FDC acceptability, allowing incorporation of multiple stakeholder perspectives.

## Supporting information

**S1 Text. Data collection tools.**
(DOCX)

**S1 Checklist. Standards for Reporting Qualitative Research (SRQR) checklist.**
(DOCX)

**S2 Checklist. Inclusivity in global research questionnaire.**
(DOCX)

**S1 Appendix. Theoretical Framework of Acceptability constructs applied to FDCs for hypertension.**
(DOCX)

## Author contributions

**Conceptualization:** Ruth Willis, Mary Gichagua, Peter Mugo, Adrianna Murphy.

**Data curation:** Daniel Mbuthia, Ruth Willis.

**Formal analysis:** Daniel Mbuthia, Ruth Willis, Mary Gichagua, Peter Mugo, Adrianna Murphy.

**Funding acquisition:** Adrianna Murphy.

**Investigation:** Daniel Mbuthia, Ruth Willis, Mary Gichagua, Peter Mugo, Adrianna Murphy.

**Methodology:** Daniel Mbuthia, Ruth Willis, Peter Mugo, Adrianna Murphy.

**Project administration:** Daniel Mbuthia, Ruth Willis, Mary Gichagua, Jacinta Nzinga, Peter Mugo, Adrianna Murphy.

**Resources:** Mary Gichagua, Peter Mugo, Adrianna Murphy.

**Software:** Daniel Mbuthia, Ruth Willis.

**Supervision:** Ruth Willis, Jacinta Nzinga, Peter Mugo, Adrianna Murphy.

**Validation:** Daniel Mbuthia, Ruth Willis, Mary Gichagua, Jacinta Nzinga, Peter Mugo, Adrianna Murphy.

**Visualization:** Daniel Mbuthia, Ruth Willis, Peter Mugo, Adrianna Murphy.

**Writing – original draft:** Daniel Mbuthia, Ruth Willis, Peter Mugo, Adrianna Murphy.

**Writing – review & editing:** Daniel Mbuthia, Ruth Willis, Mary Gichagua, Jacinta Nzinga, Adrianna Murphy.

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
