## [Decision Letter · Decision Letter 0]

15 Jul 2024

PGPH-D-24-00354

Acceptability of fixed-dose combination treatments for hypertension in Kenya: a qualitative study using the Theoretical Framework of Acceptability.

Dear Dr. Murphy,

Thank you for submitting your manuscript to PLOS Global Public Health. After careful consideration, we feel that it has merit but does not fully meet PLOS Global Public Health’s publication criteria as it currently stands. Therefore, we invite you to submit a revised version of the manuscript that addresses the points raised during the review process.

We look forward to receiving your revised manuscript.

Kind regards,

Ari Probandari, PhD

Academic Editor

Journal Requirements:

2. In the online submission form, you indicated that "The dataset used during this study is available from the authors upon reasonable request and permission from the KEMRI Wellcome Trust Research and Governance Committee". 

3. Uploaded as supplementary information.

Additional Editor Comments (if provided):

Please respond the reviewers comments and revise accordingly.

Reviewers' comments:

Reviewer's Responses to Questions

**Comments to the Author**

1. Does this manuscript meet PLOS Global Public Health’s publication criteria?

Reviewer #1: Yes

Reviewer #2: Yes

Reviewer #3: Yes

2. Has the statistical analysis been performed appropriately and rigorously?

Reviewer #1: Yes

Reviewer #2: Yes

Reviewer #3: Yes

3. Have the authors made all data underlying the findings in their manuscript fully available (please refer to the Data Availability Statement at the start of the manuscript PDF file)?

Reviewer #1: Yes

Reviewer #2: Yes

Reviewer #3: Yes

4. Is the manuscript presented in an intelligible fashion and written in standard English?

Reviewer #1: Yes

Reviewer #2: Yes

Reviewer #3: Yes

Reviewer #1: Please see my complete comments attached. Overall this is a great paper, but there are some notable gaps in context that I'm asking you to bring into the intro and discussion sections. I look forward to the second review.

Reviewer #2: The paper is well written and the findings have been presented in an appropriate manner highlighting the key concepts of the study. There are a few things that need to be addressed or changed to make the paper easier to understand.

1. In line 40-45 include the name of the county to make it easier for reference and context without having to read the entire paper.

2. There is a bit of a mix up on the methodology used or rather the title is a bit of misleading given the methodology used in the study. The aim of the study is to explore the acceptability of the FDCs among patients, caregivers and health workers. In congruence with the title line 103-104 indicates that the conceptualization is based on the Theoretical Framework of Acceptability (TFA). In line 119-120 the study design indicates that the study relied on an explorative qualitative study design that allowed for exploration and understanding of phenomenon with limited prior evidence. In line 200 - 202 it is highlighted that the study used an abductive thematic analysis approach which builds on ‘grounded theory’

approaches but explicitly allows for framing of analysis with reference to existing theory, with the aim of fostering conceptual innovation. These presents three separate methodological approaches that can be confusing to most readers. In light of this, rationalize all these methodologies in a way that they do not point to differing approaches within the same study and even possibly adjusting the title in this regard.

3. Indicate the names of the specific hospitals that were included as part of the study within the study to provide proper context for the study. This makes it easier to have proper reference for the study and where specific study participants were selected from even though the county has been indicated.

Reviewer #3: The manuscript is well written and it makes a good contribution to knowledge. However, the result presentation could be better with additional graphics and pictures for easy comprehension. The tables summarizing participant characteristics and facility details are helpful. However, additional visual aids (e.g., thematic maps or models) could be added to enhance the understanding of the findings.

**Do you want your identity to be public for this peer review?** For information about this choice, including consent withdrawal, please see our Privacy Policy

Reviewer #1: **Yes: ** Beth A Tippett Barr

Reviewer #2: No

Reviewer #3: **Yes: ** Prof. Lukman Olajide Abdur-Rahman

---

## [Decision Letter · Decision Letter 1]

1 Nov 2024

PGPH-D-24-00354R1

Acceptability of fixed-dose combination treatments for hypertension in Kenya: a qualitative study using the Theoretical Framework of Acceptability.

Dear Dr. Murphy,

Thank you for submitting your manuscript to PLOS Global Public Health. After careful consideration, we feel that it has merit but does not fully meet PLOS Global Public Health’s publication criteria as it currently stands. Therefore, we invite you to submit a revised version of the manuscript that addresses the points raised during the review process.

We look forward to receiving your revised manuscript.

Kind regards,

Ari Probandari, PhD

Academic Editor

Journal Requirements:

**Comments to the Author**

Reviewer #2: All comments have been addressed

Reviewer #4: All comments have been addressed

publication criteria?

Reviewer #2: Yes

Reviewer #4: Yes

3. Has the statistical analysis been performed appropriately and rigorously?

Reviewer #2: Yes

Reviewer #4: N/A

4. Have the authors made all data underlying the findings in their manuscript fully available (please refer to the Data Availability Statement at the start of the manuscript PDF file)?

Reviewer #2: Yes

Reviewer #4: Yes

5. Is the manuscript presented in an intelligible fashion and written in standard English?

Reviewer #2: Yes

Reviewer #4: Yes

Reviewer #2: All comments and issues addressed. I have no other issues.

Reviewer #4: The authors present a polished manuscript of acceptability of FDCs for hypertension in Kenya using the TFA. I have no comments.

**Do you want your identity to be public for this peer review?** For information about this choice, including consent withdrawal, please see our Privacy Policy

Reviewer #2: No

Reviewer #4: No

---

## [Editor Report · Decision Letter 2]

10 Jan 2025

Acceptability of fixed-dose combination treatments for hypertension in Kenya: a qualitative study using the Theoretical Framework of Acceptability.

PGPH-D-24-00354R2

Dear Dr. Murphy,

We are pleased to inform you that your manuscript 'Acceptability of fixed-dose combination treatments for hypertension in Kenya: a qualitative study using the Theoretical Framework of Acceptability.' has been provisionally accepted for publication in PLOS Global Public Health.

Best regards,

Ari Probandari, PhD

Academic Editor
